# Characterization of TCRβ and IGH Repertoires in the Spleen of Two Chicken Lines with Differential ALV-J Susceptibility Under Normal and Infection Conditions

**DOI:** 10.3390/ani15030334

**Published:** 2025-01-24

**Authors:** Meihuizi Wang, Qihong Zhang, Rongyang Ju, Junliang Xia, Chengxun Xu, Weiding Chen, Xiquan Zhang

**Affiliations:** 1State Key Laboratory of Swine and Poultry Breeding Industry, Guangdong Provincial Key Lab of Agro-Animal Genomics and Molecular Breeding, College of Animal Science, South China Agricultural University, Guangzhou 510642, China; wz@stu.scau.edu.cn (M.W.); zqh03281314@163.com (Q.Z.); ju699121@163.com (R.J.); xjl465181167@126.com (J.X.); xuchengxunscau@163.com (C.X.); cwd_ddlh@163.com (W.C.); 2Key Laboratory of Chicken Genetics, Breeding and Reproduction, Ministry of Agriculture and Rural Affair, South China Agricultural University, Guangzhou 510642, China

**Keywords:** ALV-J, differential susceptibility, antibody immune response, TCRβ repertoire, IGH repertoire, diversity

## Abstract

Some chickens are naturally more resistant to diseases than others. Avian leukosis virus subtype J (ALV-J) is a virus that harms poultry health, and in this study, we compared two chicken strains (E-line and M-line), to understand why their resistance differs. The E-line was more vulnerable to ALV-J, partly because its immune response to the virus was slower. By studying key components of the immune system, we found that the M-line had a wider variety of immune cells, allowing it to recognize more threats. After infection, the two lines adjusted their immune systems differently to fight the virus. These findings suggest that having a diverse range of immune cells helps chickens better resist ALV-J. Understanding this can help breed healthier chickens and improve poultry farming, benefiting both farmers and consumers.

## 1. Introduction

The avian leukosis virus (ALV), a member of the *Alpharetrovirus* genus within the *Retroviridae* family, is a significant pathogen affecting poultry health worldwide. It causes a variety of neoplastic diseases, as well as immunosuppression and growth retardation, leading to severe economic losses in the poultry industry [1,2]. Based on the characteristics of its viral envelope proteins, ALV is classified into 11 subgroups: A, B, C, D, E, F, G, H, I, J, and K. Among these, avian leukosis virus subgroup J (ALV-J) is particularly notorious for its high pathogenicity and impact on commercial poultry production [3]. Efforts to control ALV-J have been impeded by the lack of effective vaccines or antiviral drugs. In China, intensive purification programs have been implemented in poultry farms to reduce ALV-J prevalence, but the outcomes remain inconsistent across regions. Large-scale farms still report sporadic outbreaks of ALV-J, which may be attributed to suboptimal purification practices, the virus’s high mutational capacity, and differences in the genetic resistance of local chicken breeds [4,5]. Epidemiological studies highlight the rapid evolution of ALV-J to enhance replication and facilitate cross-infection among flock variants, underscoring its adaptive potential [6,7]. However, systematic investigations into breed-specific susceptibility to ALV-J remain limited. Genetic and immunological factors are thought to influence resistance to ALV-J. Slow-feathering chickens have shown greater susceptibility to ALV-J compared to fast-feathering chickens, possibly due to polymorphisms in the PRLR gene [8]. Similarly, studies on infectious bronchitis virus demonstrated breed-specific immune responses, with Taiwanese chickens and White Leghorn chickens exhibiting distinct gene expression profiles after vaccination [9]. In order to prevent the widespread infection of ALV-J and advance the progress of disease-resistant breeding, it is essential to gain a comprehensive understanding of the varying resistance of local chicken breeds to ALV-J.

In previous investigations, we observed that E-line chickens from a specific poultry farm in Guangdong Province exhibited a higher susceptibility to ALV-J. A statistical analysis of data collected over the past three years revealed that the incidence of avian leukosis in E-line chickens was consistently higher compared to other chicken lines. Notably, the M-line chickens from the same farm showed the most significant difference in disease incidence, with an overall prevalence rate that was only one-third of that observed in the E-line chickens. The E-line was developed at South China Agricultural University through crossbreeding between Yuehuang chickens and Isa Brown layers, whereas the M-line was selectively bred by the Department of Animal Science at South China Agricultural University to establish a distinct strain of partridge chickens. These two chicken strains exhibit different susceptibilities to ALV-J, but the underlying reasons for this phenomenon remain unclear. This study aims to investigate the potential immunological mechanisms contributing to the variation in ALV-J susceptibility between the E-line and M-line chickens. Insights gained from this research may provide a deeper understanding of the genetic and immunological factors influencing resistance or vulnerability to ALV-J infection in chickens.

The initiation of an antibody-mediated immune response begins with the engagement of the B cell receptor (BCR) on mature B cells by its specific ligand, leading to B cell activation [10]. These responses are generally classified into T cell-dependent (TD) and T cell-independent (TI) immune responses based on the type of antigen involved [11,12]. TI antigens, such as capsular polysaccharides and bacterial DNA, elicit B cell activation without the involvement of T cells [13]. In the context of ALV-J, the structural glycoprotein gp85 has been identified as its principal immunogenic determinant [14]. In TD immune responses, B cells are typically activated in secondary lymphoid organs like the spleen and lymph nodes upon recognizing protein antigens through their BCR. These activated B cells undergo extensive proliferation, class-switch recombination (CSR), and somatic hypermutation (SHM) of their immunoglobulin (Ig) gene loci, ultimately enhancing antibody affinity and specificity [15,16]. Antigen-presenting cells, such as macrophages and dendritic cells, internalize and process antigens into peptides that are presented on major histocompatibility complex (MHC) molecules. These MHC–antigen peptide complexes are recognized by T helper cells, which, through cytokine secretion and co-stimulatory signals (e.g., CD40-CD40L interaction), facilitate B cell activation and affinity maturation [17,18]. As in mammals, the adaptive immune system of chickens relies on a delicate balance between T cell-mediated cellular immunity and B cell-mediated humoral immunity. However, ALV-J has exhibited rapid mutation rates in certain local chicken populations, forming new evolutionary branches with distinct biological characteristics [4]. Such mutations enable pathogens like viruses and bacteria to evade host immune recognition, underscoring the importance of the diversity in T cell receptor (TCR) and BCR repertoires for mounting effective immune responses.

Each BCR consist of two identical heavy chains and two identical light chains, while TCR are formed as heterodimers [19]. Each T/B cell expresses only one type of TCR/BCR to recognize the corresponding antigen and rapidly proliferates and differentiates upon appropriate signaling. The diversity of BCR and TCR is primarily generated by DNA rearrangements during the early development of lymphocytes [20,21]. The diversity of TCRs and BCRs is generated through somatic V(D)J recombination during lymphocyte development, orchestrated by the RAG1-RAG2 protein complex, which initiates the rearrangement of variable (V), diversity (D), and joining (J) gene segments at antigen receptor loci [22,23]. Additional diversity arises from nucleotide insertions and deletions at recombination junctions, contributing to the complementarity-determining region 3 (CDR3), the primary site for antigen recognition [24]. Apart from the limited availability of V genes in the TCR loci, other characteristics that may affect the avian TCR repertoire include the relative simplicity of the major histocompatibility complex (MHC) and the expression of a single dominant MHC class I and II gene [25,26]. The Ig gene rearrangement in chickens has evolved not for the purpose of producing antibody diversity itself but for generating an immunoglobulin variable region that can be diversified through subsequent somatic gene conversion events [27]. The diversity of TCR and BCR is the foundation of the adaptability and effectiveness of the immune system, especially in response to rapidly mutating viruses.

The avian immune system serves as a crucial model for studying immune responses, with the TCR and BCR loci in chickens having been accurately annotated. However, comparative analyses of immune receptor repertoires across different chicken populations, particularly in the context of disease resistance, remain scarce. Immune repertoire sequencing, including both TCR and BCR sequencing, allows for a precise characterization of receptor diversity and the examination of clonal expansions, thereby providing valuable insights into the adaptive immune mechanisms of chickens. We obtained comprehensive T cell receptor beta chain (TCRβ) and immunoglobulin heavy chain (IGH) repertoires from the spleen tissues of these two chicken strains and performed sequence alignments using the IMGT database (http://www.imgt.org/ (accessed on 28 December 2024)). Our analysis focused on the comparative characterization of TCR and BCR repertoires under uninfected and ALV-J antigen-stimulated conditions in both E- and M-lines. By exploring the clonal diversity of TCR and BCR in different chicken lines, this study provides deeper insights into the mechanisms underlying chicken adaptive immune responses.

## 2. Materials and Methods

### 2.1. Ethics Statement

The animal study was reviewed and approved by South China Agriculture University’s Institutional Animal Care and Use Committee (Permit Number: 2022F140). All animal procedures were performed according to the regulations and guidelines established by this committee and the international standards for animal welfare.

### 2.2. Animal, Cell, and Virus

Two-week-old female E-line and M-line chickens were obtained from a breeder farm in Zhaoqing City, Guangdong Province, China. All the chickens have been tested and confirmed to be free of ALV-J infection prior to the experiment. The DF-1 cell line was preserved by the Key Laboratory of Chicken Genetics, Breeding, and Reproduction, Ministry of Agriculture and Rural Affairs. The SCAU-HN06 cloned strain of ALV-J was generously provided by Professor Cao Weisheng from the College of Veterinary Medicine, South China Agricultural University (Guangzhou, China).

### 2.3. ALV-J Inoculation and Sample Collection

The E-line and M-line chickens were randomly divided into control and infection groups *(n* = 16 per group), with 8 E-line and 8 M-line chickens in each group. At four weeks of age, the chickens in the infection group were inoculated intraperitoneally with 0.8 mL of ALV-J strain SCAU-HN06 (10^4^ TCID_50_/0.1 mL), while those in the control group received 0.8 mL of sterile PBS. Blood samples were collected from all groups at 7, 14, 21, and 28 days post-infection (DPI). Place the heparin-sodium anticoagulation tubes into the centrifuge and centrifuge at 4 °C, 3000 rpm for 5 min to separate the plasma. Extract the serum from the blood that has clotted in the clotting tubes. Cloacal swabs were also collected at these time points. At 21 DPI, spleens were collected from 3 ALV-J antibody-positive M-line chickens, 3 antibody-negative E-line chickens in the infection group, and 3 chickens from each line in the control group.

### 2.4. ALV-J Shedding, ALV-J Viremia, and ALV-J Antibody Were Detected by ELISA

ALV-J shedding was assessed using cloacal swabs tested for the P27 antigen according to the ALV-P27 Ag Test Kit protocol (IDEXX, Inc., Westbrook, ME, USA). Viremia was evaluated by inoculating the plasma samples into DF-1 cells at 37 °C for 7 days, followed by the detection of the P27 antigen in the cell supernatants using the same test kit. Samples with S/P ratios exceeding 0.2 were considered positive for viral presence. ALV-J antibodies in serum samples were detected using a commercial ALV-J antibody test kit (IDEXX, Inc.) as per the manufacturer’s instructions. Serum samples with S/P ratios exceeding 0.6 were considered antibody-positive.

### 2.5. RNA Isolation and Sequencing

RNA was extracted from spleen tissue using the RNeasy Plus Mini Kit (Qiagen, Hilden, Germany), and the quality of the extracted RNA was assessed using the Agilent 2100 Bioanalyzer (Beijing, China). Following quality verification, RNA was used for first-strand cDNA synthesis using SuperScript III (Life Technologies, Thermo Fisher Scientific, Carlsbad, CA, USA). Full-length TCRβ and IGH genes were amplified using a 5′RACE PCR protocol, which incorporates unique molecular barcodes (UMBs) during cDNA synthesis to minimize PCR and sequencing errors. Sequencing was performed using the ImmuHub TCR profiling system (ImmuQuad Biotech, Hangzhou, China) in PE150 mode. The 5′RACE protocol utilized a common adapter with UMBs during cDNA synthesis and reverse primers targeting constant (C) regions of IGH genes to facilitate unbiased amplification. Quality control was performed using Qubit, Agilent 2100, Agilent 4200, and qPCR. Sequencing data were processed to assess fragment content, integrity, insert size, and effective concentration.

### 2.6. Sequencing Data Analysis

The raw sequencing data underwent quality control and alignment with the IMGT database (http://www.imgt.org/ (accessed on 28 December 2024)). V, D, J, and C segments were analyzed, and CDR3 regions were extracted and assembled into clonotypes. Sequences with out-of-frame or stop codons were excluded, retaining only effectively rearranged sequences for further analysis. The clonal frequency was determined based on shared CDR3 nucleotide sequences.

### 2.7. Calculation of the Diversity Index and the Clonality Index

We assessed the clonal diversity of IGH and TCRβ repertoires using three established diversity indices: clonal richness, the Shannon–Wiener index, and the inverse Simpson index. Clonal richness is calculated by counting the number of all the clones. The Shannon index is used to measure the diversity of TCR/BCR clones in a sample. A higher Shannon index indicates a higher diversity of clones. The index is calculated using the following formula: $H^{\prime }=-\sum_{i}^{n}pi(\ln pi)$; where $H^{\prime }$ represents the Shannon–Wiener index, n is the total number of unique clones in the repertoire, and pi is the relative abundance of the $i^{th}$ clone, calculated as the ratio of the number of sequences belonging to that clone to the total number of sequences in the repertoire. The inverse Simpson index, which is the reciprocal of Simpson’s index of diversity, is used to gauge clonal diversity with a focus on clone evenness. The formula for the inverse Simpson index is $D_{inv}=\frac{1}{D}=\frac{1}{\sum_{i=1}^{N}P_{i}^{2}}$; where $D_{inv}$ is the inverse Simpson index, D is Simpson’s index of diversity, N is the total number of sequences in the repertoire, and $p_{i}$ is the number of sequences of the $i^{th}$ clone. The dose of effect for 50% (DE50) is an index used to measure clonality. The calculation involves ranking the clone sequences by frequency in descending order and summing the frequencies from the highest until their cumulative total accounts for 50% of the overall clone frequency. The proportion of clone types corresponding to this cumulative frequency represents the DE50. A lower DE50 value suggests a higher clonality, indicating that specific clones have undergone significant expansion.

### 2.8. Statistical Analysis

Statistical analyses were conducted using SPSS 23.0 and GraphPad Prism 8.0.1. Data are presented as the mean ± standard error of the mean (SEM). Group comparisons were performed using independent samples *t*-tests and a one-way ANOVA. Statistical significance was defined at *p* < 0.05, with thresholds for high significance at *p* < 0.01 and *p* < 0.001.

## 3. Results

### 3.1. Detection of ALV-J Shedding, ALV-J Viremia, and ALV-J Antibody

The dynamics of ALV-J shedding, viremia, and antibody responses were monitored in the E-line and M-line chickens for 28 days post-infection (DPI). In the infected group, the S/P values of the P27 antigen in cloacal swabs from some E-line chickens exceeded 0.2 at all the time points (7, 14, 21, and 28 DPI), whereas only three M-line chickens exhibited ALV-J shedding at 14 DPI. Significant differences in S/P values between the two lines were observed at 21 and 28 DPI (Figure 1A). The positivity rate for ALV-J viremia was consistently higher in the E-line than in the M-line during the first three time points, with a significant difference noted at 14 DPI (Figure 1B). These findings align with prior reports of avian leukosis virus infection in chicken farms. The rate of antibody production in response to ALV-J infection differed markedly between the two lines. By 14 DPI, one M-line chicken had developed ALV-J antibodies, whereas the E-line chickens did not produce detectable antibodies until 28 DPI. Significant differences in antibody S/P values between the two lines were observed at 21 DPI (Figure 1C).

### 3.2. Features of the Sequence Data

At 21 DPI, three chickens from the infected M-line produced detectable ALV-J antibodies, whereas all the chickens in the infected E-line remained negative for ALV-J antibodies, with the statistical analysis confirming a significant difference between the two groups (Figure 1C). To further explore the observed discrepancy in immune responses, spleens were harvested at 21 DPI from three chickens per strain in both the control and infected groups for the TCR-seq and BCR-seq analysis. Among the chickens selected at this time point, all the infected M-line chickens had detectable ALV-J antibodies, while the E-line did not. The average number of analyzed TCRβ sequences was 13,225 for the control E-line, 36,295 for the control M-line, 38,372 for the infected E-line, and 45,640 for the infected M-line. For the IGH repertoire, the average sequence numbers were 115,674 (control E-line), 119,738 (control M-line), 384,547 (infected E-line), and 289,473 (infected M-line).

### 3.3. TCRβ Repertoire Diversity

We first conducted a comprehensive analysis of TCRβ repertoire diversity in the spleen. As illustrated in Figure 2, the evaluation of three diversity metrics did not reveal any significant disparities in TCRβ diversity between the control M-line and the infected M-line. In contrast, a substantial enhancement in TCRβ diversity was noted in the infected E-line relative to its control group. This enhancement is corroborated by the increased count of T cell receptor beta locus (TRB) clonotypes in the infected E-line, as compared to its control group (Figure 2A). Moreover, the inverse Simpson index indicated a notable increment in the diversity of high-frequency clones within the infected E-line (Figure 2C). Nonetheless, the infected E-line did not exceed the infected M-line in terms of TCRβ diversity, with no statistically significant differences observed. Regarding the overall TCRβ diversity, the M-line within the control cohort exhibited a superior clonal diversity compared to the E-line (Figure 2B). The findings imply that the M-line sustains a higher degree of TCRβ repertoire diversity irrespective of ALV-J exposure, while the E-line demonstrated a significant increment in clonotype richness and the diversity of high-frequency clonotypes subsequent to infection.

### 3.4. TRB Clonal Homeostasis

A bubble plot depicting TRB clonotypes showed a higher abundance of clonotypes in the infected group compared to controls (Figure 3A). Ranking the clonotypes by abundance revealed that the top 20 TRB clones contributed a significantly higher proportion in the infected M-line compared to the control M-line (Figure 3B). Specific clone amplification was stronger in infected groups but did not reach statistical significance (Figure 3C). The public clones of TRB between the E- and M-lines were limited, constituting less than 20% of the total clonotypes in both groups. Among the public clones, the proportion in the E-line infection group was significantly higher than in the other three groups (*p* < 0.01) (Figure 3D). Conversely, the proportion of private TRB clones in the E-line infection group was significantly lower than in the other three groups .

### 3.5. Consistent Patterns in the CDR3 Length Distribution and V/J Gene Segment Usage Are Observed Between the E-Line and M-Line TCRβ Repertoires

The length distributions of complementarity-determining region 3 (CDR3) within the TCRβ repertoires of the E- and M-line chickens were similar and unaffected by ALV-J infection (Figure 4A). In the two chicken strains, the detected V gene segments involved in TRB rearrangement include TRBV1 and TRBV2, while the J gene segments include TRBJ1, TRBJ3, TRBJ4, and TRBJ5. The usage of TRBV and TRBJ genes in the control groups of the E- and M-lines was similar (Figure 4B). Following infection with ALV-J, the E- and M-line chickens exhibited distinct patterns of TRBJ gene usage (Figure 4C). In the E-line, there was a decrease in the usage of the TRBJ4 and an increase in the usage of the TRBJ5. Infection with ALV-J did not induce any changes in the TRBJ gene usage in the M-line chickens (Figure 4D), although an increased usage of TRBV1 and a decreased usage of TRBV2 were observed compared to the control M-line. However, these differences did not reach statistical significance. TCRβ V-J pairings were consistent across all groups, with TRBV1/TRBJ5 being the most frequent combination (13.7–25.0%) and TRBV2/TRBJ4 the least frequent (5.2–7.5%) (Figure 4E).

### 3.6. IGH Repertoire Diversity

The diversity of the IGH repertoire was assessed across groups (Figure 5). The clonal richness of the infected M-line was significantly higher than that of the control M-line (Figure 5A). No significant differences were observed in the diversity of high-frequency clones, as measured by the inverse Simpson index (Figure 5C). Within the infection groups, the M-line exhibited a significantly higher clonal diversity (Shannon index) compared to the E-line, while no significant differences were found between the E- and M-lines in the control group (Figure 5B). Additionally, no significant differences were observed in the diversity of high-frequency clones, as measured by the inverse Simpson index (Figure 5C).

### 3.7. IGH Clonal Homeostasis

The bubble plot depicts the number of IGH clones and their respective frequencies across the samples, with the clonotypes more abundant in the infected M-line (Figure 6A). Ranking the clones by abundance revealed comparable proportions across different categories among groups (Figure 6B). However, the infected M-line exhibited a significantly greater clonal expansion than the control M-line (Figure 6C). Following ALV-J infection, both lines had a stronger clone amplification compared to their respective control groups, although the differences did not reach statistical significance. The public clones of IGH between the E- and M-lines were limited, accounting for less than 20% of the total clonotypes in both lines (Figure 6D). Among the public clones, the proportion in the E-line infection group was significantly higher than in the control groups of both the E- and M-lines, but not significantly different from the JM group. Conversely, the proportion of private clones in the JE group was significantly lower than in the control groups of both lines.

### 3.8. Similarity and Heterogeneity of the CDR3 Length Distribution and V/J Gene Segment Usage of the IGH Repertoire in Two Strains of Chickens

The length distribution of IGH CDR3 sequences was consistent across the control groups for both the E- and M-lines, as observed in Figure 7A. Similarly, within the E-line, no significant differences in the CDR3 length distribution were detected between the infected and uninfected groups, as also illustrated in Figure 7B. In contrast, the infected M-line had a significantly higher proportion of IGH CDR3 sequences exceeding 85 bp and a lower proportion within the 45–55 bp range compared to the control M-line (Figure 7C). We identified a total of 81 V segments involved in IGH gene rearrangement. In the control group, significant differences in usage rates were observed for IGHV1-1, IGHV1-16, IGHV1-19, IGHV1-31, IGHV1-41, IGHV1-56, and IGHV1-61-1 between the E- and M-line (Figure 7D). The infection group and the control group of the E-line showed significant differences in the usage of IGHV1-10, IGHV1-14, IGHV1-21, and IGHV1-72 (Figure 7E). ALV-J infection led to an increase in the usage of high-frequency V segments such as IGHV1-1, IGHV1-34, and IGHV1S1 in the E-line, although these increases were not statistically significant. Similarly, the infection group and the control group of the M-line displayed significant differences in the usage of IGHV1-10, IGHV1-13, IGHV1-61-1, IGHV1-77, and IGHV1S2 (Figure 7F). Despite the increased usage of high-frequency V segments such as IGHV1-1, IGHV1-53, and IGHV1S1 in the M-line due to ALV-J infection, these changes did not reach statistical significance.

### 3.9. The Phenomena of Class-Switch Recombination (CSR) and Somatic Hypermutation (SHM) in the Immune System

Class-switch recombination (CSR) and somatic hypermutation (SHM) are two crucial mechanisms that B cells employ to enhance the effectiveness of the adaptive immune response. The frequency of CSR in the infected M-line was significantly higher than in the control M-line (Figure 8A). The infected E-line also showed an increase in CSR frequency compared to its control, although this was lower than that of the infected M-line and was not statistically significant. The frequencies of SHM showed that the mutation frequency of the IGHY gene in the E- and M-lines of the infected group increased, while the mutation frequencies of the IGHM and IGHA genes decreased (Figure 8B); these changes did not reach statistical significance.

## 4. Discussion

Despite the implementation of continuous biosecurity measures in poultry farms, the complete eradication of ALV remains a significant challenge for the poultry industry. ALV infections, especially those caused by ALV-J, are prevalent among diverse local chicken breeds in China, often characterized by mixed infections within subpopulations [6]. The persistence and adaptability of ALV-J are attributed to its high mutation rate, exemplified by mutations such as N123I, which enhance the virus’s binding to the chNHE1 receptor, thereby facilitating replication and transmission [5]. Beyond genetic engineering, other strategies like selective breeding for disease resistance, vaccination, and antiviral therapeutics offer additional avenues for combating ALV-J [28,29,30,31]. These strategies necessitate an intensive examination of host immune factors, as well as a detailed exploration of the underlying mechanisms governing the host immune system.

In our study, the E-line chickens exhibited a higher susceptibility to ALV-J, as evidenced by the higher viremia and virus shedding rates under identical experimental conditions. The weaker antibody-mediated immune responses in the E-line chickens, compared to the M-line, suggest that differences in humoral immunity may underlie their greater susceptibility. This finding aligns with previous studies that suggest the influence of genetic background on immune response profiles to viral infections [8,9]. These differences emphasize the need for further research into immune system variability across different chicken breeds, which may inform breeding strategies for improved disease resistance.

As in mammals, the adaptive immunity (or specific immunity) of poultry relies on the structure of the receptor repertoires that recognize antigens. T lymphocytes and B lymphocytes with different receptor types are important components in the functioning of the immune response [32]. High-throughput sequencing allows for a comprehensive analysis of the TCR and BCR repertoire characteristics, providing insights into the diversity of TCR and BCR repertoires and clonal expansions under specific conditions. This is an important method for analyzing the structure of and changes in T and B lymphocyte populations. Based on the ALV-J antibody detection results, we selected the spleens from E-line and M-line chickens during the period of pronounced antibody level discrepancies for TCR-seq and BCR-seq analysis. We focused on the spleen tissue of chickens, as the spleen is a crucial secondary lymphoid organ involved in immune responses. Notably, a high proportion of rare clones has been observed in the spleens of all bird species [33]. Certainly, the characteristics of immunoreceptor repertoires in other immune organs also await exploration.

We compared the TCRβ and IGH repertoires between the E-line and M-line control groups, which represent the baseline immune repertoires of the two chicken lines prior to antigen exposure. Clone diversity metrics revealed a higher level of diversity in the TCRβ complexes within the M-line as compared to the E-line, while there are no significant differences in the clonal homeostasis of the TCRβ repertoire and in the analysis of CDR3 sequences. For the IGH repertoire, our data show that, apart from the significant differences in the usage of 7 IGHV gene segments, there were no other differences between the control group E- and M-lines. The generation of a diverse BCR or TCR is the basis of adaptive immunity, and the TCR diversity index in healthy donors decreases with age [34]. The diversity of the initial immune repertoire is beneficial for recognizing a wide variety of antigens. Based on previous studies, the TCRβ repertoire of the M-line exhibits a higher diversity, which may correspond to a stronger ability to resist ALV-J invasion. In addition, the homeostasis of the TCRβ and IGH repertoires in the control groups is relatively consistent between the two chicken lines, which may arise from conserved regulatory mechanisms in chickens in the absence of pathogen invasion. This is consistent with the lack of significant differences in TCR diversity observed in some local chicken breeds [35].

To understand the unique immune response characteristics of each line, we compared the control and infected groups within the same line. The sequence data analysis revealed that the infection groups had a higher number of TRB and IGH sequences compared to the control groups, indicating that T cells and B cells were in an activated state in the spleen tissues of both lines following ALV-J infection. In terms of immune repertoire diversity, the infected group of the E-line showed a higher clonality richness and diversity of high-frequency clones in the TCRβ repertoire compared to its control group, whereas the infected group of the M-line exhibited no significant differences. In contrast, for the IGH repertoire, the infected group of the M-line exhibited a significantly higher IGH repertoire diversity compared to its control group, whereas the E-line did not show such differences. Both E-line and M-line IGH and TCRβ repertoires exhibited specific changes in response to ALV-J antigen stimulation. However, the distinct changes observed post-infection were primarily in the IGH repertoire, with fewer changes in the TCRβ repertoire. For instance, regarding CDR3 length distribution, the M-line infection group showed a significantly higher proportion of IGH CDR3 sequences exceeding 85 bp compared to the control M-line, while the proportion within the 45-55 bp range was lower. Regarding V/J gene usage, the E-line infection group exhibited significant differences in the usage of IGHV1-10, IGHV1-14, IGHV1-21, and IGHV1-72 compared to its control group, whereas the M-line infection group displayed significant differences in the usage of IGHV1-10, IGHV1-13, IGHV1-61-1, IGHV1-77, and IGHV1S2 in comparison to its control group. Nevertheless, no significant differences were observed between the two lines and their respective control groups in terms of TRB CDR3 length distribution, as well as the usage of TRBV and TRBJ genes. To understand the unique immune response characteristics of each line, we defined clones shared by two or more individuals within each group as public clones, while clones unique to each individual in the group were considered private clones. If different individuals independently produce the same antibody in response to an antigen, it could create a consistent and collective selective pressure on that particular epitope, increasing the likelihood of the emergence of escape variants at that site [36,37]. Due to the significant proportion of public clones in the infected group of the E-line, it may suggest that the E-line has produced a more general immune response to ALV-J, whereas individuals in the infected group of the M-line may have generated more unique immune responses to ALV-J.

In the human body, it has been observed that there are differences in the average CDR3 length between the peripheral blood mononuclear cells (PBMCs) of patients infected with the highly pathogenic avian influenza virus H5N6 and the control group. Specifically, the usage rates of TRBV12-3, TRBV12-4, and TRBV15 gene segments were higher in the H5N6-infected group [38]. TCRβ clone amplification has been detected in both colonized and germ-free birds, and the intergroup differences suggest the influence of the microbiota [39]. Therefore, the contributions of CDR3 sequences of varying lengths and the functional domains encoded by specific V and J segments may tend to select ALV-J antigen-specific clones and promote their expansion. The differences in IGH clone CDR3 diversity and the biased usage of IGHV genes between the two chicken strains in this study may imply different immune response strategies of the B cell repertoire to ALV-J virus infection, which in turn affects the immunization efficacy of the E-line chickens to ALV-J. During the development of the immunoglobulin CDR3 repertoire in chickens’ small intestines, the CDR3 length of IgM shortens with increasing age, while the IgA CDR3 repertoire exhibits considerable conservation during development [40]. The CDR3 length may affect the shape of the antigen recognition region, thereby influencing its binding specificity [41,42]. In this study, both chicken lines exhibited homogeneity in the CDR3 length distribution of the TCRβ repertoire and the usage of TRBV and TRBJ genes, regardless of ALV-J antigen stimulation, suggesting the possible existence of a specific conservative mechanism.

Some local chicken breeds possess rich immune genetic resources, and the analysis of V-J pairing and CDR3 diversity in four local chicken breeds showed no significant differences in TCR diversity [35]. Consistent with the insignificant differences in the TCRβ repertoire between the two chicken strains in this study, the specific genetic background requires further investigation. The chicken TCRα and β loci are relatively simple, with only two V gene subfamilies in each locus [43], compared to the 20-30 Vα and Vβ gene subfamilies found in mice and humans [44]. This lower number significantly limits the potential for chicken T cell receptor repertoire diversity. In various types of tumors, a higher TCR repertoire diversity is associated with a more highly activated tumor immune microenvironment (TIME), while BCR diversity is more related to antibody responses [45]. Additionally, the chicken TCRβ locus has only 4 Jβ segments, whereas mice have 12 and humans have 13. These genetic structural features result in the limited potential for TCRβ CDR3 diversity in chickens, which may partially explain the phenomenon of minimal specific differences in CDR3 regions between chicken strains. Compared with wild-type mice, mouse models expressing human terminal deoxynucleotidyl transferase (TdT) have longer N regions and more diverse CDR3 sequences, including shorter CDR3 sequences [46]. Research on TCR and BCR has largely focused on the prediction and adjuvant treatment of human cancers and tumors. TCR clonality was increased in T cells from peripheral blood in advanced HCC, compared with the early and middle stages [47]. Currently, the combination of single-cell sequencing and TCR-seq/BCR-seq has been applied in humans and mice, with single-cell RNA and BCR sequencing performed on lymph nodes, spleens, and lung tissues from mice infected with influenza, identifying several germinal-center (GC) B cell subpopulations and organ-specific differences [48].

The clonal diversity of IGH in the two chicken strains is higher than the clonal diversity of TCRβ, which may be related to the unique mechanisms of IGH diversification. IGH diversification mechanisms include not only V(D)J recombination, non-template nucleotide insertion between fragments by RAG enzymes, and light chain to heavy chain pairing, but also somatic hypermutation (SHM). SHM and CSR diversify immunoglobulin (lg) genes and are initiated by the activation induced deaminase [49]. SERCA proteins modulate intracellular Ca^2+^ levels to regulate RAG1 and RAG2 gene expression and V(D)J recombination and defects in SERCA functions cause lymphopenia [48,50]. Template insertion has also been identified as a fourth mechanism contributing to BCR repertoire diversity, with RAG-l mediated V(D)J recombination rather than AID activity likely being the most plausible mechanism for template insertion in the CDR3 region [51]. However, the role of template insertion in the immune system is not yet clear. However, the majority of CDR3 diversity arises from the non-templated nucleotide insertions facilitated by the nuclear enzyme TdT, and nucleotide deletions primarily occur at V(D)J junctions due to the action of exonucleases [52]. We have conducted a preliminary exploration of the immune repertoire characteristics in two chicken lines. In future studies, we plan to conduct large-scale analyses of the changes in TCR and BCR repertoires during immune reactivation across diverse chicken genetic backgrounds, and to further investigate the mechanisms underlying the formation of chicken-specific CDR3 region diversity.

## 5. Conclusions

In summary, this study indicates that the diversity of the TCRβ repertoire and the IGH repertoire’s response to ALV-J differ among chicken strains with varying susceptibility to ALV-J. Under normal conditions, the chicken line more susceptible to ALV-J (E-line) exhibits lower TRB diversity. Upon ALV-J infection, the diversity of the IGH repertoire significantly increases in the anti-ALV-J infected chicken line (M-line), whereas no such change is observed in E-line. Moreover, the IGH clone expansion in chicken lines with different ALV-J sensitivities also shows distinct preferences, such as changes in CDR3 length and V segment usage, which could affect the efficiency of antigen recognition against ALV-J. During ALV-J infection, TRB clone variations are similar across chicken lines with different susceptibilities to ALV-J.

Therefore, differences in immune receptor diversity may underlie the varying susceptibility of different chicken strains to ALV-J, with more resistant strains potentially possessing a superior immune receptor repertoire. This study is the first to compare the changes in TCRβ and IGH in the spleens of normal and ALV-J infected chickens, which is beneficial for identifying immune molecular targets for chicken viruses. The specificity of immune repertoires in different chicken strains warrants further investigation.

## Figures and Tables

**Figure 1 animals-15-00334-f001:**
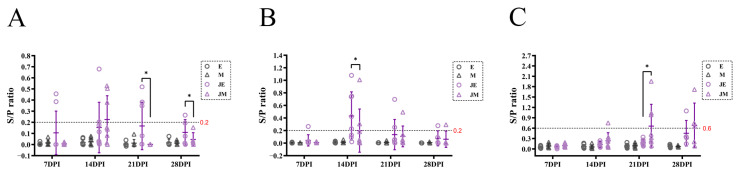
Compared to M-line, E-line is more susceptible to ALV-J and shows slower efficiency in producing ALV-J antibodies. Groups in the figure: E (control E-line), M (control M-line), JE (infected E-line), and JM (infected Line M). (**A**) ALV-J shedding was monitored on days 7, 14, 21, and 28 post-infection with the ALV-J strain SCAU-HN06, with an S/P ratio greater than 0.2 (red) indicating a positive result for the anal p27 antigen; (**B**) ALV-J viremia was monitored, with an S/P ratio greater than 0.2 (red) indicating a positive result for ALV-J viremia; (**C**) ALV-J antibody level, with an S/P ratio greater than 0.6 (red) indicating a positive result for ALV-J antibodies. Statistically significant differences between the model estimates are depicted above the plots based on their corresponding *p*-values: * *p* < 0.05.

**Figure 2 animals-15-00334-f002:**
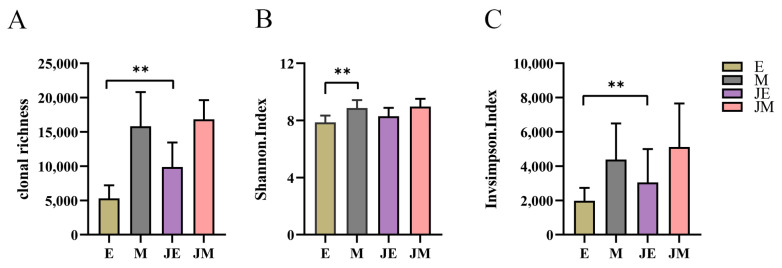
TCRβ repertoire diversity. The designations E, M, JE, and JM correspond to specific groupings as illustrated in Figure 1. (**A**) The clonal richness of TCRβ; (**B**) Shannon index reflects overall TCRβ clonotype diversity, with higher values indicating greater diversity; and (**C**) inverse Simpson index represents the diversity of high-frequency clonotypes in the TCRβ repertoire, with larger values indicating higher diversity. Statistically significant differences between the model estimates are depicted above the plots based on their corresponding *p*-values: ** *p* < 0.01.

**Figure 3 animals-15-00334-f003:**
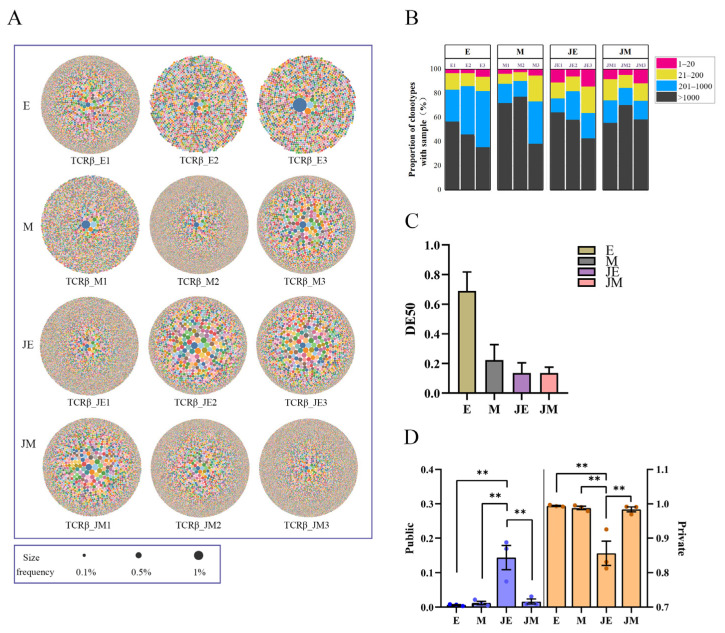
TRB clonotype homeostasis exhibits distinct patterns of change in Lines E and M following ALV-J infection. The designations E, M, JE, and JM correspond to specific groupings as illustrated in Figure 1. (**A**) Bubble plots represent the frequency of each clonotype. The more bubbles and the denser they are, the greater the number of that clonotype in the sample. The bubble at the center denotes the predominant clonotype, and its size is proportional to its frequency; (**B**) the proportion of TRB clones with different frequency abundances. Each bar of three bars for the E, M, JE, and JM lines corresponds to the respective three samples for each line. Clonotype abundance is defined as the frequency of each clonotype relative to the total sequence count. Clonotypes are ranked by abundance and divided into four categories: the top 20 most abundant (red), ranks 21–200 (yellow), ranks 201–1000 (blue), and ranks > 1000 (black); (**C**) DE50 values are utilized to quantify the clonality of the TCRβ repertoire. Higher DE50 values indicate a more uniform distribution of clonotypes, with decreasing DE50 values signifying increased clonality, which suggests the expansion of specific clones; (**D**) proportions of private and public clonotypes in the TCRβ repertoire. Clonotypes shared among two or more samples within the same group are classified as public clonotypes, whereas those unique to individual samples are identified as private clonotypes. Statistically significant differences between the model estimates are depicted above the plots based on their corresponding *p*-values: ** *p* < 0.01.

**Figure 4 animals-15-00334-f004:**
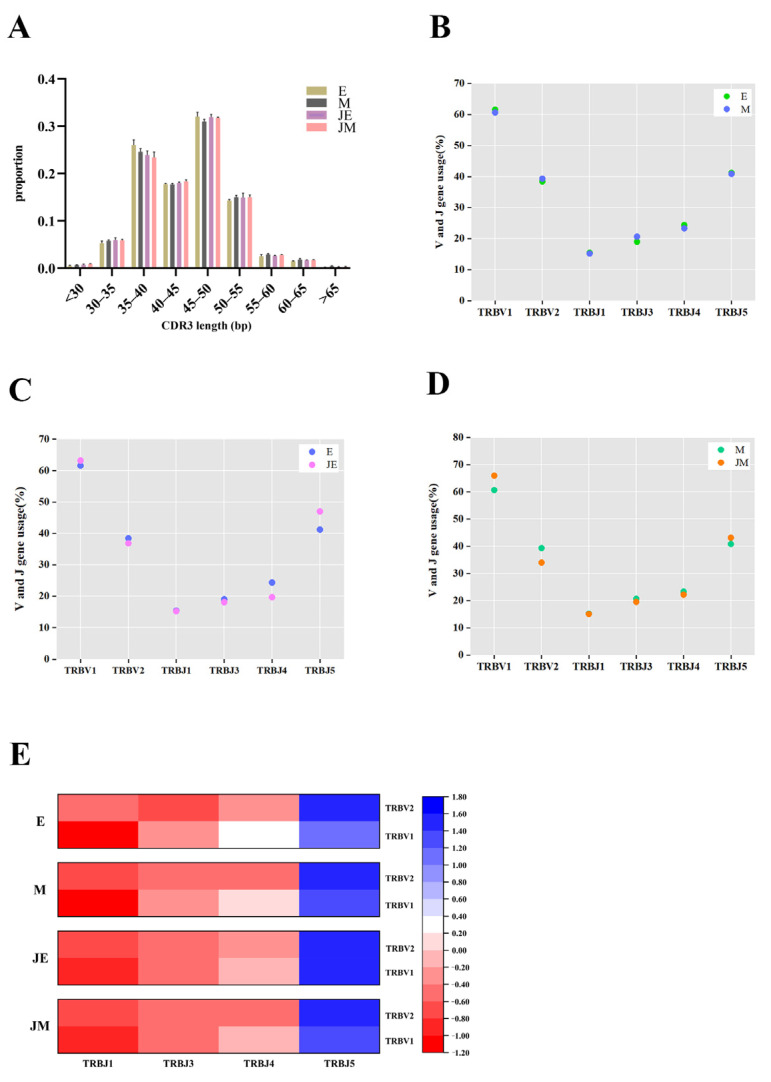
The TCRβ repertoires of E- and M-lines showed no differences in CDR3 length distribution or V/J gene segment usage. The designations E, M, JE, and JM correspond to specific groupings as illustrated in Figure 1. (**A**) Distribution of CDR3 sequences with different lengths; (**B**) usage frequencies of V/J gene segments: comparison between control E-line and control M-line; (**C**) usage frequencies of V/J gene segments: comparison between control E-line and infected E-line; and (**D**) usage frequencies of V/J gene segments: comparison between control M-line and infected M-line. (**E**) Heatmap of V-J pairings in the TCRβ repertoire.

**Figure 5 animals-15-00334-f005:**
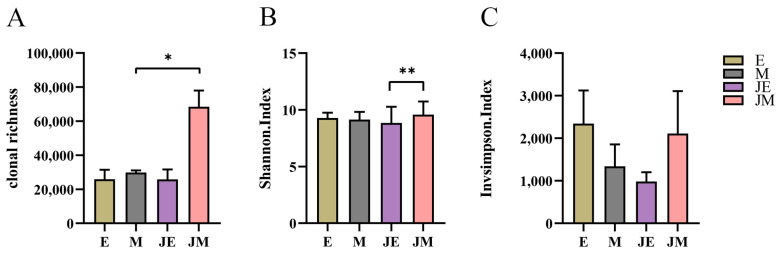
ALV-J infection increased IGH clonotype richness in M-line, with a higher IGH clonotype diversity observed in the infected M-line compared to the infected E-line. The designations E, M, JE, and JM correspond to specific groupings as illustrated in Figure 1. (**A**) IGH clonotype richness is represented by the number of IGH clonotypes. (**B**) Shannon index reflects overall IGH clonotype diversity, with higher values indicating greater diversity; (**C**) inverse Simpson index represents the diversity of high-frequency clonotypes in the IGH repertoire, with larger values indicating higher diversity. Statistically significant differences between the model estimates are depicted above the plots based on their corresponding *p*-values: * *p* < 0.05; ** *p* < 0.01.

**Figure 6 animals-15-00334-f006:**
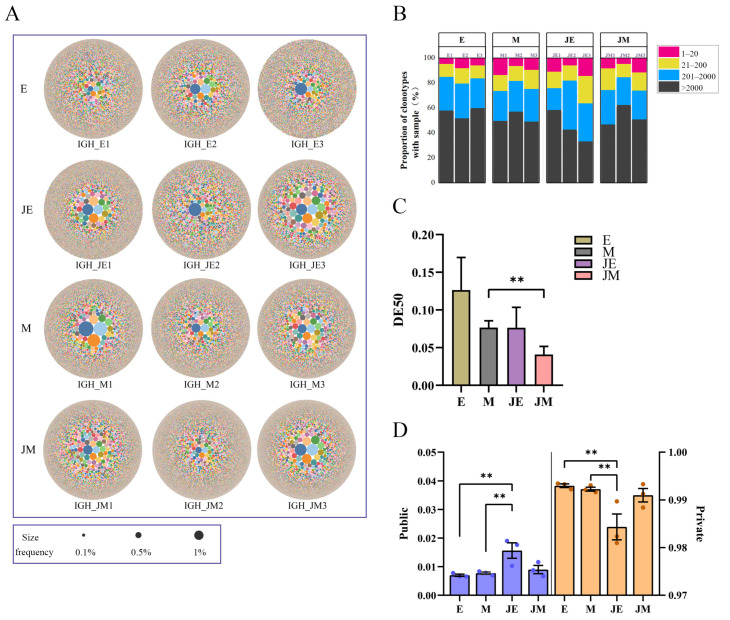
IGH clonotype homeostasis exhibited distinct patterns of change in Lines E and M following ALV-J infection. The designations E, M, JE, and JM correspond to specific groupings as illustrated in Figure 1. (**A**) Bubble plots represent the frequency of each clonotype. The more bubbles and the denser they are, the greater the number of that clonotype in the sample. The bubble at the center denotes the predominant clonotype, and its size is proportional to its frequency; (**B**) the proportion of IGH clones with different frequency abundances. Each bar of three bars for the E, M, JE, and JM lines corresponds to the respective three samples for each line. Clonotype abundance is defined as the frequency of each clonotype relative to the total sequence count. Clonotypes are ranked by abundance and divided into four categories: the top 20 most abundant (red), ranks 21–200 (yellow), ranks 201–2000 (blue), and ranks > 2000 (black); (**C**) DE50 values are utilized to quantify the clonality of the IGH repertoire; and (**D**) proportions of private and public clonotypes in the IGH repertoire. Statistically significant differences between the model estimates are depicted above the plots based on their corresponding *p*-values: ** *p* < 0.01.

**Figure 7 animals-15-00334-f007:**
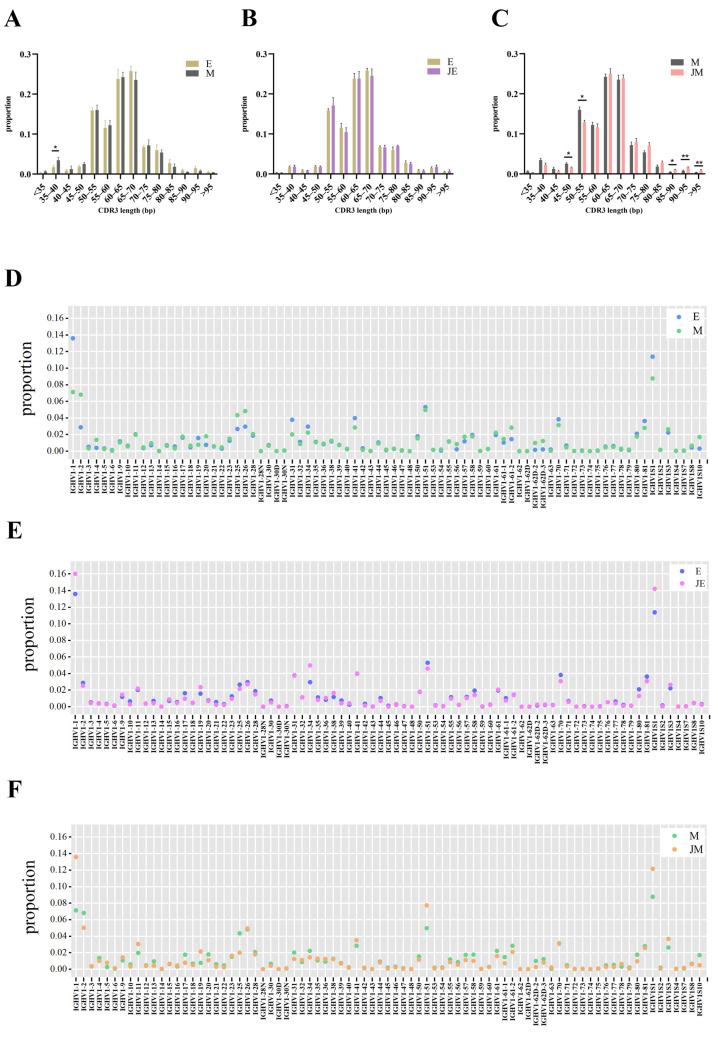
The distribution of CDR3 length and V/J gene segment usage in IGH repertoires of lines E and M. The designations E, M, JE, and JM correspond to specific groupings as illustrated in Figure 1. (**A**) CDR3 length distribution comparison: control E-line and control M-line; (**B**) CDR3 length distribution comparison: control E-line and infected E-line; (**C**) CDR3 length distribution comparison: control M-line and infected M-line; (**D**) IGH V gene segment usage frequency comparison: control E-line and control M-line; (**E**) IGH V gene segment usage frequency comparison: control E-line and infected E-line; and (**F**) IGH V gene segment usage frequency comparison: control M-line and infected M-line. Statistically significant differences between the model estimates are depicted above the plots based on their corresponding *p*-values: * *p* < 0.05; ** *p* < 0.01.

**Figure 8 animals-15-00334-f008:**
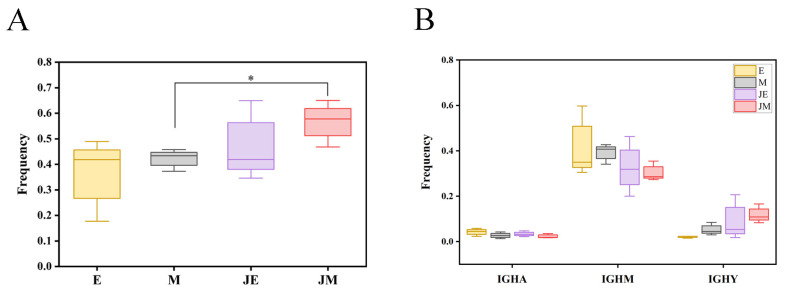
Changes in IGH class-switch recombination and somatic hypermutation frequencies in E- and M-lines following ALV-J infection. The designations E, M, JE, and JM correspond to specific groupings as illustrated in Figure 1. (**A**) Frequency of class-switch recombination. (**B**) Somatic hypermutation frequencies of three IGH subtypes. Statistically significant differences between the model estimates are depicted above the plots based on their corresponding *p*-values: * *p* < 0.05.

## Data Availability

The data presented in this study are available on request from the corresponding author. The data are not publicly available due to privacy restrictions and the long extension of datasets.

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
