# Peer review of "Characterization of TCRβ and IGH Repertoires in the Spleen of Two Chicken Lines with Differential ALV-J Susceptibility Under Normal and Infection Conditions"

_animals, 2025, doi:10.3390/ani15030334_

Round 1
Reviewer 1 Report
Comments and Suggestions for Authors
This manuscript can bring some interesting for understanding the resistance of host animals against pathogens and practical application, but there are many points for improvement in experimental design, methods, results, and writing. Some revisions are suggested to be done.
1. Please add some statements on the susceptibility of two different chicken strains (E line and M line ) against ALV in Introduction or Discussions.
2. In animal infection tests, ALV-J shedding, ALV-J viremia and ALV-J antibody in in uninfected chickens (Control group) should be detected and the data should be added in Figure 1.
3. In Figure 2 and Figure 5, the degree of variation in some indicators is too high and the sample numbers are too little (N=3) so that it is difficult to determine the significance of the difference between groups.
4. The manuscript should add some more information on E-line and M-line infected chickens collected for high-throughput detection in Methods, define whether the chickens carried ALV-J or not.
5. The author should further improve the quality of some figures. For an example, in Figure 7C-E, the letters on the horizontal axis are too blurry and small to show their names.
6. Some misspelling or non-standard language expressions should be revised, for some examples many TCRB in Discussions, and P<0.05 or 0.01 in Results.
Reviewer 2 Report
Comments and Suggestions for Authors
Major question 1
It is stated that the formula for Shannon Index is 𝐻′ = −∑ 𝑝𝑖 (ln 𝑝𝑖), and the formula for inverse Simpson’s index is 𝐷_𝑖𝑛𝑣 = 1/𝐷 = 1/ (∑ _ (𝑖 = 1) ^𝑁 𝑃_𝑖^2). From lines 187-189, the authors state that “In both equations, the term ‘pi’ denotes the frequency of a particular clone type, while ‘N’ denotes the total number of clones. DE50 as an indicator of clonality (degree of cloning) measures the degree of homogeneity of clonality.”
Authors should clearly define the other parameters, such as 𝑖, 𝑃_𝑖. Where is DE50 from? Is it the same as 𝐷? It should be clearly explained for a better understanding of the calculation of the diversity and clonality index.
Major question 2
From lines 228 – 231, the authors state, “To investigate the differential immune responses, spleens were collected from control and infected groups of both lines at 21 DPI for TCR-seq and BCR-seq analysis. Among the chickens selected at this time point, all infected M-line chickens had detectable ALV-J antibodies, while E-line did not.”
The authors should clearly state why they had investigated differential immune responses at 21 DPI, seeing that in Figure 1C, the E line was negative for ALV-J antibodies at 21 DPI.
Major question 3
The explanation of the results of the TCRβ repertoire diversity, lines 236-247, should be extensively revised and clearly written, directing the readers to the correct figures.
Major question 4
Looking at Figure 3B, it is difficult to understand how the statements made from lines 259-261 were reached.
Authors should indicate what the 3 different bars for the E, JE, M, and JM lines in Figure 3B represent. Figure 3A is not self-explanatory; the authors should add a legend.
Major question 5
The authors made some assumptions in lines 459 – 461; this should be backed with appropriate citations.
In lines 462 – 464, the authors state, “We also found that the differences in CDR3 length distribution and V/J gene usage induced by the ALV-J antigen were concentrated in the IGH repertoire, not in the TCRβ repertoire, between the two lines”. The statement should be further discussed and backed with appropriate citations.
If the statement in lines 480-483 is a general statement from literature, it should be backed with appropriate citations. If it is from the findings of this study, the authors should refer the readers to the figures or charts supporting this statement.
Generally, the discussion part should be rewritten by the authors. The results should be discussed sequentially following the results of this study. In writing the discussion, the authors should refer the readers to some of the relevant figures of the results mentioned in the discussion; this would make it easier for the readers to comprehend.
Minor question 1
In lines 240 - 241, the authors referred to the Inverse Simpson Index as Figure 2A instead of Figure 2C; it should be carefully checked and corrected.
Minor question 2
In line 244, the authors should verify if they are referring to Figure 2C or 2B.
Minor question 3
Figure 4A, the authors should add a title to the X-axis
Minor question 4
Figure 6A is not self-explanatory; the authors should add a figure legend. What each of the bars for E, JE, M, and JM lines represents in Figure 6B should be explained by the authors, the same as in Figure 3B.
Minor question 5
In line 353, no differences should be changed to no significant differences.
Minor question 6
The authors should add a title to the X-axis in Figures 7A and 7B. Figures 7C – 7E are too small to be seen, making it difficult to follow the explanation of the results. The authors should enlarge the figures.
Minor question 7
The authors should add a figure legend in Figure 8B.
In line 388, Figure 8B should be in bold font to ensure consistency.
Generally, the same pattern should be followed in the arrangement of the figures to ensure consistency and to facilitate easy understanding of the results. Some were E, M, JE, JM, and others were E, JE, M, JM. Authors should consider following a similar pattern for all figures.
Minor question 8
In line 477, compositional should be changed to composition.
Comments on the Quality of English LanguageThe authors should consider comprehensive English language editing by native speakers to ensure sentence coherence and consistency in the tenses.
Author Response
Please see the attachment,

Reviewer 3 Report
Comments and Suggestions for Authors
In the manuscript ” Characterization of TCRβ and IGH repertoires in the spleen of two chicken lines with differential ALV-J susceptibility under normal and infection”, The authors investigated and compared two chicken strains, Line E and Line M, in their susceptibility to the avian leukosis virus subgroup J (ALV-J). The manuscript is generally well addressed and well -written. However, I have some comments/suggestions.
Line 13: Simple Summary, please follow the journal guidelines using abstract only.
Line 13: “Avian leucosis virus subtype J….” please correct the word leucosis.
Line 151: the sentence “Blood samples were collected from all groups at 7,14,21, and 28 days post-infection 151 (DPI). Plasma was separated by centrifugation at 3,000 rpm for 5 minutes” please rewrite (line 148-152) to avoid repetition.
Line 154: “At 21 DPI, spleens were collected from 3 ALV-J antibody-positive M-line chickens…....... “ that means at 28 DPI, the number of birds decreased to 5, correct? if so, do you think 5 samples per line are enough to show significance from statistical prospective?
Line 167: the sentence “RNA was extracted from spleen tissue…” . please provide the kits used for extraction.
Line 557: References: All references need to be revised following the journal guidelines. References # 4, 8, 9, 26, 29, 34, 35, 36, 40, 43, 44, 46, 47, 48, are incomplete. please revise adding missing info. Reference #6 is incorrect, please revise.
Round 2
Reviewer 1 Report
Comments and Suggestions for Authors
The Manuscript have been revised according to my suggestion and have been improved greatly. I agreed to be accepted.
Reviewer 2 Report
Comments and Suggestions for Authors
I have no further comments